

# Identification of key genes and biological processes contributing to colitis associated dysplasia in ulcerative colitis

Di Zhang[1,2], Pengguang Yan[1,2], Taotao Han[1,2], Xiaoyun Cheng[1,2] and Jingnan Li[1,2]

[1] Department of Gastroenterology, Peking Union Medical College Hospital, Peking Union Medical College, Chinese Academy of Medical Sciences, Beijing, China
[2] Key Laboratory of Gut Microbiota Translational Medicine Research, Chinese Academy of Medical Sciences, Beijing, China

## ABSTRACT

**Background.** Ulcerative colitis-associated colorectal cancer (UC-CRC) is a life-threatening complication of ulcerative colitis (UC). The mechanisms underlying UC-CRC remain to be elucidated. The purpose of this study was to explore the key genes and biological processes contributing to colitis-associated dysplasia (CAD) or carcinogenesis in UC via database mining, thus offering opportunities for early prediction and intervention of UC-CRC.

**Methods.** Microarray datasets (GSE47908 and GSE87466) were downloaded from Gene Expression Omnibus (GEO). Differentially expressed genes (DEGs) between groups of GSE47908 were identified using the "limma" R package. Weighted gene co-expression network analysis (WGCNA) based on DEGs between the CAD and control groups was conducted subsequently. Functional enrichment analysis was performed, and hub genes of selected modules were identified using the "clusterProfiler" R package. Single-gene gene set enrichment analysis (GSEA) was conducted to predict significant biological processes and pathways associated with the specified gene.

**Results.** Six functional modules were identified based on 4929 DEGs. Green and blue modules were selected because of their consistent correlation with UC and CAD, and the highest correlation coefficient with the progress of UC-associated carcinogenesis. Functional enrichment analysis revealed that genes of these two modules were significantly enriched in biological processes, including mitochondrial dysfunction, cell-cell junction, and immune responses. However, GSEA based on differential expression analysis between sporadic colorectal cancer (CRC) and normal controls from The Cancer Genome Atlas (TCGA) indicated that mitochondrial dysfunction may not be the major carcinogenic mechanism underlying sporadic CRC. Thirteen hub genes (*SLC25A3*, *ACO2*, *AIFM1*, *ATP5A1*, *DLD*, *TFE3*, *UQCRC1*, *ADIPOR2*, *SLC35D1*, *TOR1AIP1*, *PRR5L*, *ATOX1*, and *DTX3*) were identified. Their expression trends were validated in UC patients of GSE87466, and their potential carcinogenic effects in UC were supported by their known functions and other relevant studies reported in the literature. Single-gene GSEA indicated that biological processes and Kyoto Encyclopedia of Genes and Genomes (KEGG) pathways related to angiogenesis and immune response were positively correlated with the upregulation of *TFE3*, whereas those related to mitochondrial function and energy metabolism were negatively correlated with the upregulation of *TFE3*.

Corresponding author
Jingnan Li, pumcjnl@126.com

**Conclusions**. Using WGCNA, this study found two gene modules that were significantly correlated with CAD, of which 13 hub genes were identified as the potential key genes. The critical biological processes in which the genes of these two modules were significantly enriched include mitochondrial dysfunction, cell-cell junction, and immune responses. *TFE3*, a transcription factor related to mitochondrial function and cancers, may play a central role in UC-associated carcinogenesis.

## INTRODUCTION

Ulcerative colitis (UC) is a subtype of inflammatory bowel disease (IBD), which is characterized by long-standing and recurring chronic inflammation of the colonic mucosa. UC-associated colorectal cancer (UC-CRC) is the most severe and life-threatening complication, especially in patients suffering from UC for a long duration. Epidemiological data have revealed that the risk of UC-CRC increases with disease duration (*Bernstein et al., 2001*; *Bopanna et al., 2017*). Although more attention has been given to the colonoscopic screening of dysplasia and cancer in UC patients (*Laine et al., 2015*; *Rubin et al., 2019*), early diagnosis is difficult because of the flat appearance and multifocal lesions. Therefore, there is an urgent need to explore the molecular biological mechanisms underlying UC-associated carcinogenesis and to find new strategies for early diagnosis, treatment, and prevention of UC-CRC.

Early genetic changes in precancerous lesions have been reported to contribute to the initiation of carcinogenesis. For example, *Ju et al. (2020)* revealed that 70% of miRNA alterations occur during the transition from normal to a preneoplastic stage of breast cancer. Similarly, *Zhang et al. (2020)* found that most of the differentially expressed genes (DEGs) identified in high-grade intraepithelial neoplasia (HGIN) and early gastric cancer (EGC) compared to their paired controls have already changed in low-grade intraepithelial neoplasia (LGIN) lesions. In addition, they identified 22 coDEGs (co-up DEGs and co-down DEGs), which are thought to play crucial roles in gastric tumorigenesis and progression, during the stages of LGIN, HGIN, EGC, and gastric adenocarcinoma. This phenomenon offers the possibility of early diagnosis and treatment of cancers in patients with UC. In fact, some abnormal molecular events have also been demonstrated in the inflamed colonic mucosa of UC before any apparent histological evidence of colitis-associated dysplasia (CAD) or cancer (*Itzkowitz, 2006*; *Itzkowitz & Yio, 2004*; *Tang et al., 2012*). However, comprehensive analysis based on gene expression profiles to reveal the genetic changes that occur in UC and contribute to carcinogenesis is still lacking.

In the present study, using database mining, we explored the key genes and biological processes contributing to CAD that were dysregulated in UC. We conducted weighted gene co-expression network analysis (WGCNA) based on DEGs between CAD and control

groups and found two gene modules correlated with the progression of UC-associated carcinogenesis. Functional enrichment analysis of the genes in these two modules revealed the associated crucial biological processes. Thirteen hub genes were identified as the potential key genes. Furthermore, we investigated the changes in their expression in the UC and CAD groups.

## MATERIALS & METHODS

### Data selection and processing

By searching the Gene Expression Omnibus (GEO, https://www.ncbi.nlm.nih.gov/geo/) database, the microarray dataset GSE47908 based on the GPL570 platform was selected for analysis because it embraces the transcriptional profiles of colonic mucosa from both UC and CAD patients (*Bjerrum et al., 2014*). Specifically, it included data from 45 patients with UC (20 with left-sided colitis, 19 with pancolitis, and six with UC-associated dysplasia) and 15 healthy controls. After obtaining the data, principal component analysis (PCA) was performed to check and visualize the grouped data. For validation, another microarray dataset GSE87466 comprising the gene expression profiles of colonic mucosa from 87 patients with UC and 21 healthy controls was selected to check the gene expression trends in UC (*Li et al., 2018*). For comparison, we downloaded the transcriptome and clinical data of 480 patients with sporadic colorectal cancer (CRC) and 41 healthy controls from The Cancer Genome Atlas (TCGA) database (date limit: February 23, 2021), using "TCGAbiolinks" R package version 2.16.4. The gene expression profiles were normalized using the voom function in "limma" R package (*Ritchie et al., 2015*).

### Differential expression analysis

Differential expression analysis was performed using the "limma" R package (*Ritchie et al., 2015*). The sum of the mean value of absolute $\log_2$ fold change (FC) and the two standard deviations of absolute $\log_2$ FC was used as the cut-off value of $\log_2$ FC. Genes with absolute $\log_2$FC $>\log_2$ FC cut-off and adjusted $p$ value $<0.05$, were considered as DEGs. The results are visualized as volcano plots.

### WGCNA

The expression profiles of DEGs between CAD and control group were extracted to construct weighted gene co-expression network using R package "WGCNA" (*Langfelder & Horvath, 2008*; *Zhang & Horvath, 2005*). First, the candidate soft-thresholding powers (1 to 30) were used to calculate the scale independence and mean connectivity using the pickSoftThreshold function. The first candidate power, whose degree of independence was $>0.8$, was selected as the proper power. Then, a co-expression network was constructed and modules were identified using the blockwiseModules function, with the parameters mergeCutHeight set to 0.28 and minModuleSize set to 30. Each module was assigned a unique color. Finally, the Pearson correlation coefficient and corresponding $p$ value between each module's eigengene and phenotype were calculated. In addition to the UC and CAD phenotypes, another phenotype "Progress" was also included to quantify the dynamic process of UC-associated carcinogenesis. The phenotype "Progress" was assigned

the values of 0,1, and 2, to represent the control, UC, and CAD groups, respectively; this reflected the progressive process from normal to UC and then to CAD. It means that if a module is significantly correlated to "Progress", then this module is considered to be correlated to the progressive process from normal to CAD.

### Identification of hub genes

For a specified module gene, the module membership (MM) calculated by signedKME function represents its degree of importance in the module, and gene significance (GS) calculated by the cor function represents the degree of correlation with the phenotype. By calculating MM and GS, the module genes with high MM and GS can be defined as hub genes. In the present study, thresholds of MM >0.9 and GS >0.7, were chosen to screen for the hub genes of each module (*Zou et al., 2019*).

### Function enrichment analysis

Functional enrichment analysis was performed using the "clusterProfiler" R package (*Yu et al., 2012*). Significantly enriched Gene Ontology-Biological processes (GO-BP) and Kyoto Encyclopedia of Genes and Genomes (KEGG) pathways were identified using enrichGO and enrichKEGG functions, respectively. Gene set enrichment analysis (GSEA) was performed based on the gene list sorted by $log_2FC$ obtained from differential expression analysis using gseGO and gseKEGG functions (*Subramanian et al., 2005*). Single-gene GSEA was conducted based on the gene list sorted by Spearman correlation coefficient between every gene and the specified hub gene to predict the significant biological processes and pathways associated with the hub gene. A *p* value <0.05 was considered significant.

## RESULTS

### GEO data overview

A flow chart of the present study is shown in Fig. 1A. As for GSE47908 dataset, 15 controls, 20 patients with left-sided colitis, and six patients with UC-associated dysplasia were enrolled in the present study, representing the control, UC, and CAD groups, respectively. Clinical data including sex, age, disease duration, Mayo score, Mayo endoscopic score, smoking status, and daily medication were extracted and compared using the chi-squared test (*Bjerrum et al., 2014*). Results showed that there were no significant differences in sex, disease duration, smoking status, or daily medication between the groups. As for age, Mayo score, and Mayo endoscopic score, only median and interquartile ranges (IQR) were known, and the corresponding *p*-values could not be calculated (Table S1). After processing, four outliers were removed and a total of 37 samples, including 13 controls, 18 patients with UC, and six patients with CAD, were enrolled. As shown in the PCA plot (Fig. 1B), the gene transcriptional profiles of these three groups were clearly distinct from each other. The PCA plot of gene expression profiles of 87 patients with UC and 21 healthy controls in GSE87466 dataset shown in Fig. 1C, also indicates a significant distinction between these two groups.

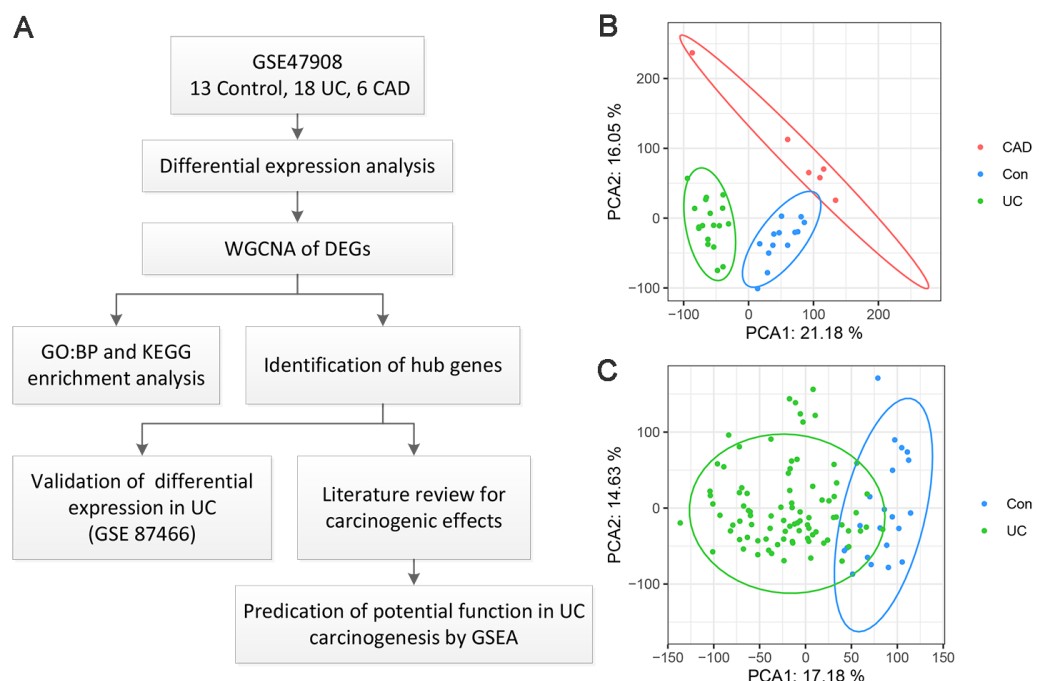

**Figure 1** **Overview of the work flow and GEO data.** (A) Flow chart of the present study; (B) Principal component analysis plot of the gene expression profiles in GSE47908. Two principal components containing 37.23% of the variance illustrate separate clustering of the three group of samples; (C) Principal component analysis plot of the gene expression profiles in GSE87466. It shows the separate clustering of the two groups with a few samples overlapping by two principal component containing 31.81% of the variance.

## Differential expression analysis

To investigate the genetic changes during the progress from normal to UC and then to CAD, differential expression analyses between UC and control, CAD and UC, and CAD and control groups in GSE47908 were performed. The results are presented in Fig. 2. There were 679 upregulated and 288 downregulated genes in UC compared to those in the control group (Fig. 2A, $\log_2$ FC cut-off of 1.092), 260 upregulated and 616 downregulated genes in CAD compared to those in the UC group (Fig. 2B, $\log_2$ FC cut-off of 1.073), and 305 upregulated and 462 downregulated genes in CAD compared to those in the control group (Fig. 2C, $\log_2$ FC cut-off of 0.804). Venn diagrams were then used to further detail the genetic changes during this dynamic process. There were 67 co-up DEGs (Fig. 2D) and 46 co-down DEGs (Fig. 2E) in UC vs. controls and CAD vs. controls. In addition, the Venn diagram shows that among 260 upregulated and 616 downregulated genes in CAD vs. UC, only 61 upregulated (Fig. 2D) and 180 downregulated genes (Fig. 2E) changed significantly compared to the control group. This indicates that some DEGs dysregulated during the progress from normal to UC may continue to change in CAD and even contribute to the initiation of CAD, while others might be only responsible for the development of UC and may change back to the normal level if the inflammation is controlled.

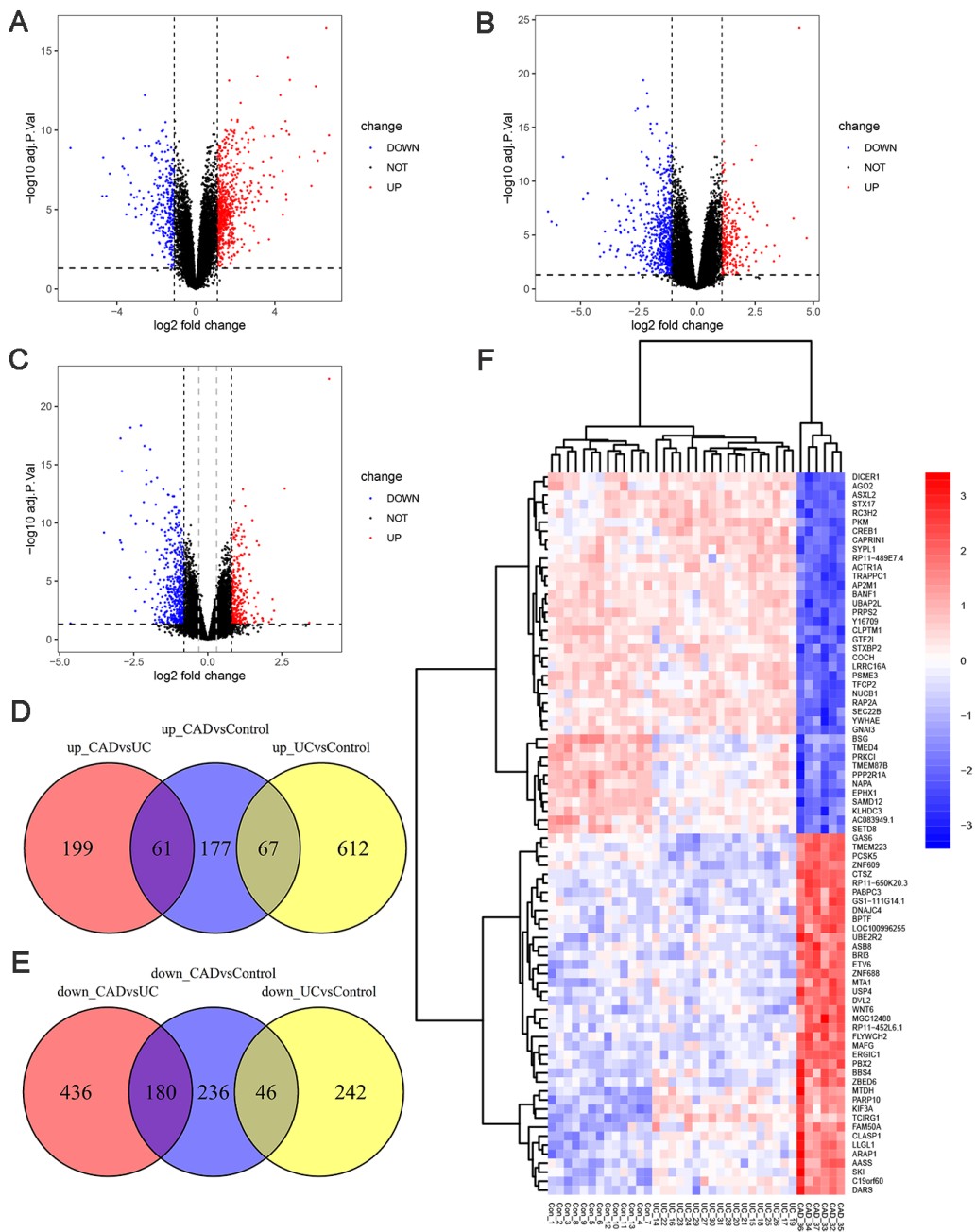

**Figure 2** **Differential expression analysis.** (A) Volcano plot between UC and control groups. There are 679 up-regulated genes and 288 down-regulated genes in UC compared to controls when $\log_2$FC cut-off value was set to 1.092 by calculated. (B) Volcano plot between CAD and UC groups. There are 260 up-regulated genes and 616 down-regulated genes in CAD compared to UC when $\log_2$FC cut-off value was set to 1.073 by calculated. (C) Volcano plot between CAD and control groups. Black vertical line indicates $\log_2$FC of 0.804, and grey vertical indicates $\log_2$FC of 0.3. There are 305 up-regulated genes and 462 down-regulated genes in CAD compared to controls when $\log_2$FC cut-off value was set to 0.804, 2904 up-regulated genes and 2025 down-regulated genes when $\log_2$FC cut-off value was set to 0.3. (D) Venn diagram illustrating up-regulated genes in different groups. (E) Venn diagram illustrating down-regulated genes in different groups. (F) Heat map of top 40 up-regulated and 40 down-regulated DEGs between CAD and controls after sorted by adjusted *p*-value from smallest to largest.

To gain further insights into the expression pattern of DEGs during the progress from normal to UC and then to CAD, the top 40 upregulated and top 40 downregulated DEGs between CAD and controls after sorting by adjusted *p*-value were visualized in a heat map (Fig. 2F). Row cluster analysis showed that these DEGs were grouped under two clusters: upregulated and downregulated genes in CAD compared to those in controls. However, when UC samples were taken into consideration, every cluster could be further divided into two sub-clusters, showing different changing trends of these DEGs during the progress from normal to CAD.

It should be noted that no co-up or co-down DEGs in UC vs. control groups and CAD vs. UC groups were identified in Venn diagrams, while the heatmap showed marked continuous changes in the expression patterns of some DEGs from control to UC and then to CAD. The traditional filter method for DEGs can potentially miss somekey information. To make our analysis more comprehensive, the cut-off criteria of $\log_2$ FC was set to 0.3, leading to the identification of 4929 DEGs between the CAD and control groups, which is a suitable number of DEGs for WGCNA analysis.

## WGCNA

WGCNA was performed using 4929 DEGs between the CAD and control groups. By setting the soft threshold to 12 (Figs. 3A–3B), a total of six functional modules, apart from the gray module, were identified (Fig. 3C). The turquoise, red, and blue modules were positively correlated with CAD, and the brown, yellow, and green modules were negatively correlated with CAD. Furthermore, the green and blue modules were correlated with UC and CAD in the same direction respectively, and were found to be correlated with "Progress" with the highest correlation coefficients (green module: correlation coefficient = −0.75, *p*-value = 9e−08; blue module: correlation coefficient = 0.71, *p*-value = 9e−07) (Fig. 3D). This indicates that genes belonging to blue and green modules and the relevant biological processes begin to change at an earlier stage and continue to change throughout the whole process of progression from normal to UC and then to CAD.

## Function enrichment analysis

The GO-BP enrichment analysis of each module is shown in Table S2. Green module genes were significantly enriched in biological processes related to mitochondrial function and energy metabolism (Fig. 4A). Consistent with this observation, KEGG pathway enrichment analysis indicated that the green module genes were significantly enriched in energy metabolism-related pathways, such as the TCA cycle, carbon metabolism, and oxidative phosphorylation (Fig. 4B). Blue module genes were significantly enriched in biological processes related to cell–cell junctions and immune responses, especially neutrophil-mediated immunity (Fig. 4C). The KEGG pathways that blue module genes were significantly enriched in included infection-related pathways, focal adhesion, tight junction, Hippo signaling pathway, Notch signaling pathway, and ErbB signaling pathway (Fig. 4D).

To further explain the role/s of mitochondrial dysfunction during the process of UC-associated carcinogenesis, GSEA was performed to analyze the alterations in

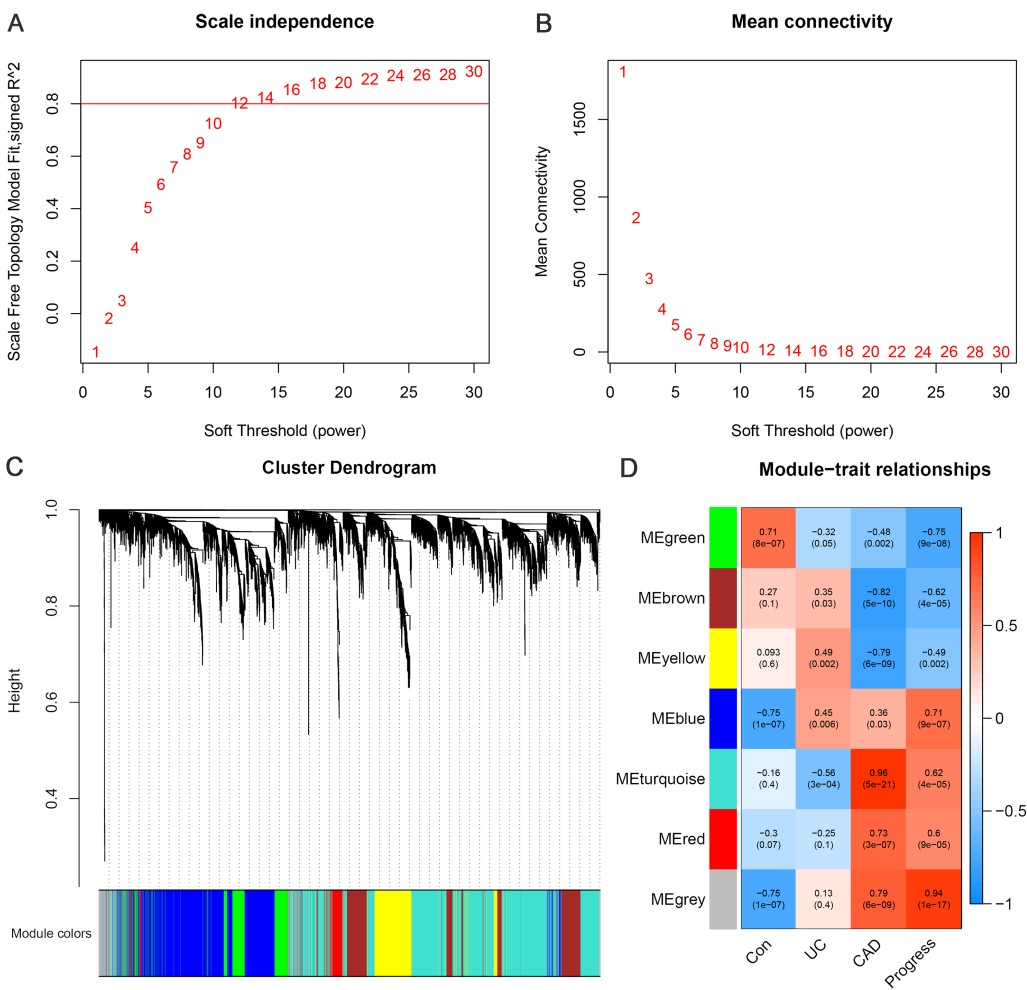

**Figure 3  Weighted gene co-expression network analysis.** (A) Network topology for different soft-thresholding powers. The scale-free topology can be attained at the soft-thresholding power of 12. (B) Cluster dendrogram and gene modules identified by WGCNA. (C) Correlation heat map of gene modules and traits, labelled with correlation coefficient and *p* value. The trait "Progress" means the progressive process from normal to UC, and then to CAD, which is assigned the value of 0, 1 and 2, to represent the control, UC and CAD group, respectively.

mitochondrial function-related GO-BP in the UC and CAD groups. The results indicated that mitochondrial function-related GO-BP, including cellular respiration, oxidative phosphorylation, respiratory electron transport chain, and mitochondrial gene expression, were all significantly downregulated in UC (Fig. S1A) and CAD (Fig. S1B) groups as compared to those observed for the control group.

To explore the differences between UC-CRC and sporadic CRC in terms of the alteration of mitochondrial functions, the gene expression profiles of 480 sporadic CRC patients and 41 healthy controls from the TCGA database were analyzed. After differential expression analysis, GSEA of GO-BP was conducted based on the gene list sorted by $\log_2$ FC. The results showed that biological processes, including mitochondrial gene expression and

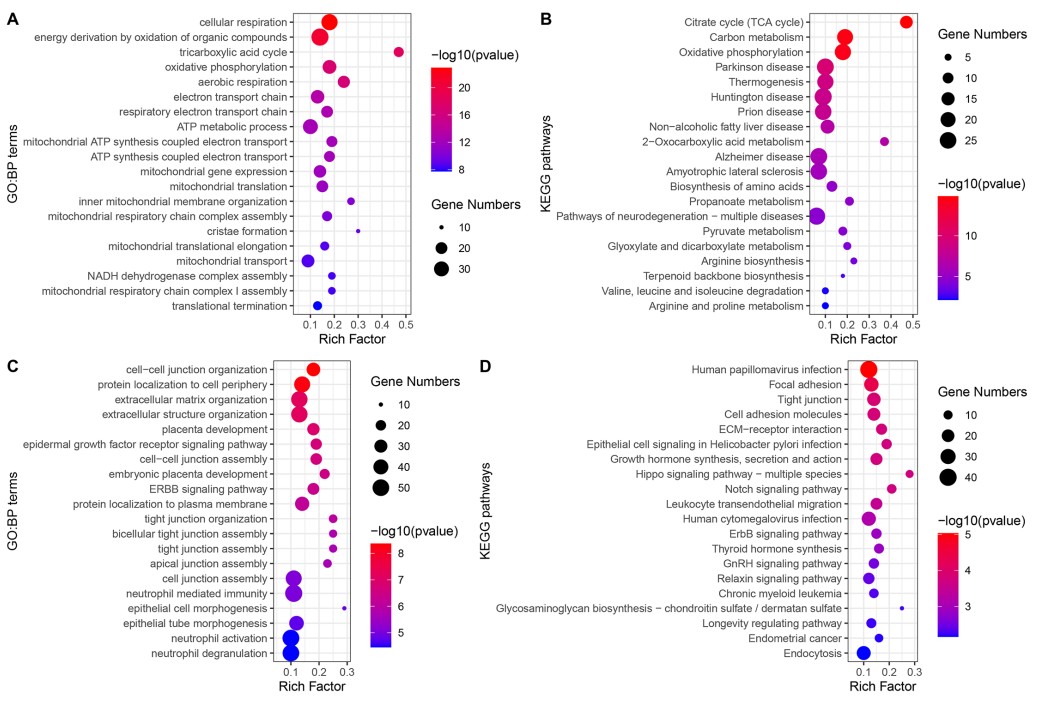

**Figure 4 GO-BP and KEGG pathways enrichment analysis.** (A) Biological processes enriched in green module genes. (B) KEGG pathways enriched in green module genes. (C) Biological processes enriched in blue module genes. (D) KEGG pathways enriched in blue module genes.

mitochondrial translation, were upregulated in sporadic CRC patients, and no other mitochondrial function-related biological processes were identified (Table S3). This indicates that the carcinogenic mechanism underlying sporadic CRC is different from that of UC-CRC.

## Identification and validation of hub genes

Based on MM >0.9 and GS >0.7, seven genes (*SLC25A3*, *ACO2*, *AIFM1*, *ATP5A1*, *DLD*, *TFE3*, and *UQCRC1*) were identified as hub genes in the green module (Fig. 5A) and six hub genes (*ADIPOR2*, *SLC35D1*, *TOR1AIP1*, *PRR5L*, *ATOX1*, and *DTX3*) were identified in the blue module (Fig. 5C). The differential expression patterns of these 13 hub genes during the progression from normal to UC and then to CAD are displayed in Figs. 5B and 5D. All of them were observed to be continuously up/downregulated significantly, supporting their potential persistent roles in UC-associated carcinogenesis.

Considering the small sample size of the GSE47908 dataset, validation using another dataset was considered indispensable. To validate the changes in the inflammatory phase, differential expression analysis of the 13 hub genes was performed in UC patients from GSE87466 dataset (87 patients with UC and 21 controls). The results showed that the expression changes of hub genes in UC patients of GSE87466 dataset were consistent with those observed for patients belonging to GSE47908 dataset (Fig. 6). To further explore the potential association of hub genes with UC-associated carcinogenesis, a brief literature review was performed to search for the proof of their carcinogenic effects (*Bruggemann et*

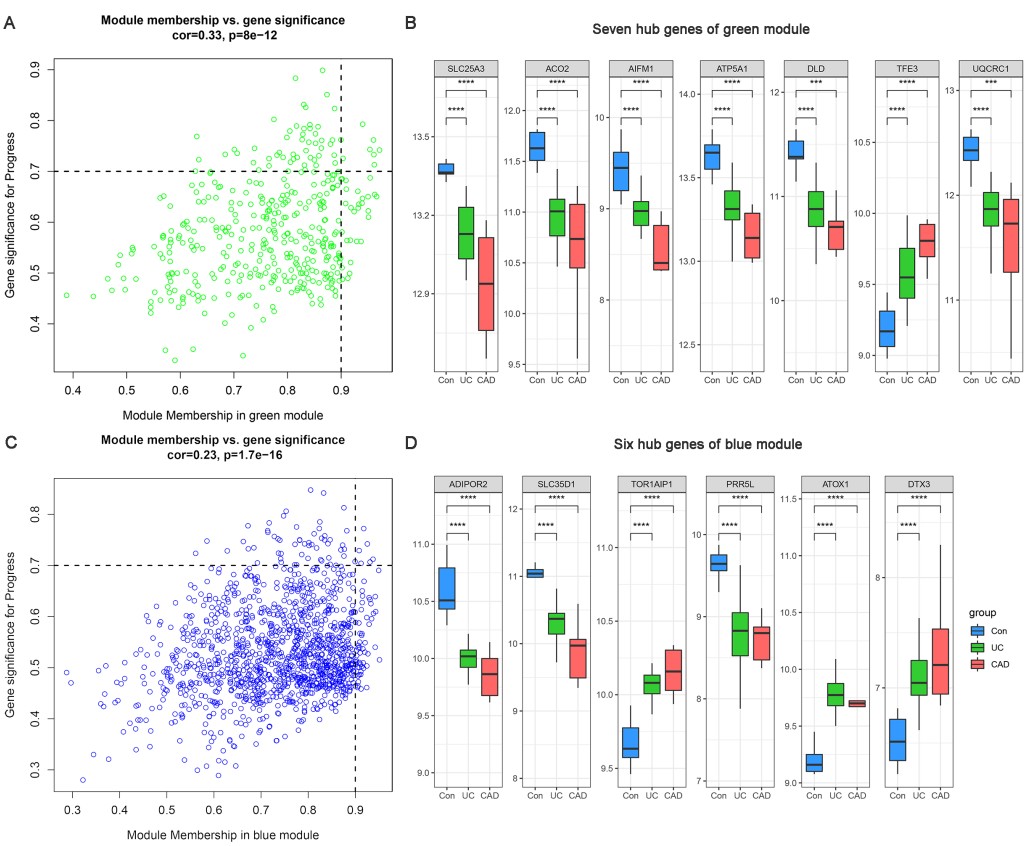

**Figure 5  Hub genes of green and blue modules.** (A) Identification of hub genes of green module by MM and GS filtering. (B) Differential expression patterns of seven hub genes of green module. (C) Identification of hub genes of blue module by MM and GS filtering. (D) Differential expression patterns of six hub genes of blue module.

*al., 2017*; *Byeon et al., 2010*; *Chung et al., 2017*; *Ciccarone et al., 2020*; *Ding et al., 2020*; *Jana et al., 2020*; *Karginova et al., 2019*; *Khalil, 2007*; *Kim et al., 2019*; *Laiho et al., 2003*; *Li et al., 2019*; *Linehan et al., 2019*; *Liu et al., 2018*; *Millan & Huerta, 2009*; *Oehler et al., 2009*; *Perera et al., 2015*; *Rao et al., 2019*; *Seth et al., 2009*; *Thedieck et al., 2007*; *Viola et al., 2017*; *Wang et al., 2013*; *Wang et al., 2020*). The results are presented in Table 1. With the exception of *TOR1AIP1* and *PRR5L*, almost all other hub genes have been reported to be involved in cancers. Furthermore, five of the hub genes (*AIFM1*, *ATP5A1*, *UQCRC1*, *ADIPOR2*, and *ATOX1*) may play roles in the carcinogenesis of sporadic CRC. These known associations between hub genes and cancers suggest the possible carcinogenic effects of these hub genes in CAD.

## Predication of the potential function of TFE3 in UC-associated carcinogenesis by GSEA

Among the proteins encoded by the hub genes, TFE3, a transcription factor, has been known to be involved in the onset and progress of cancers by regulating many biological processes, such as energy metabolism, lysosomal biogenesis, and immune response (*Beckmann,*

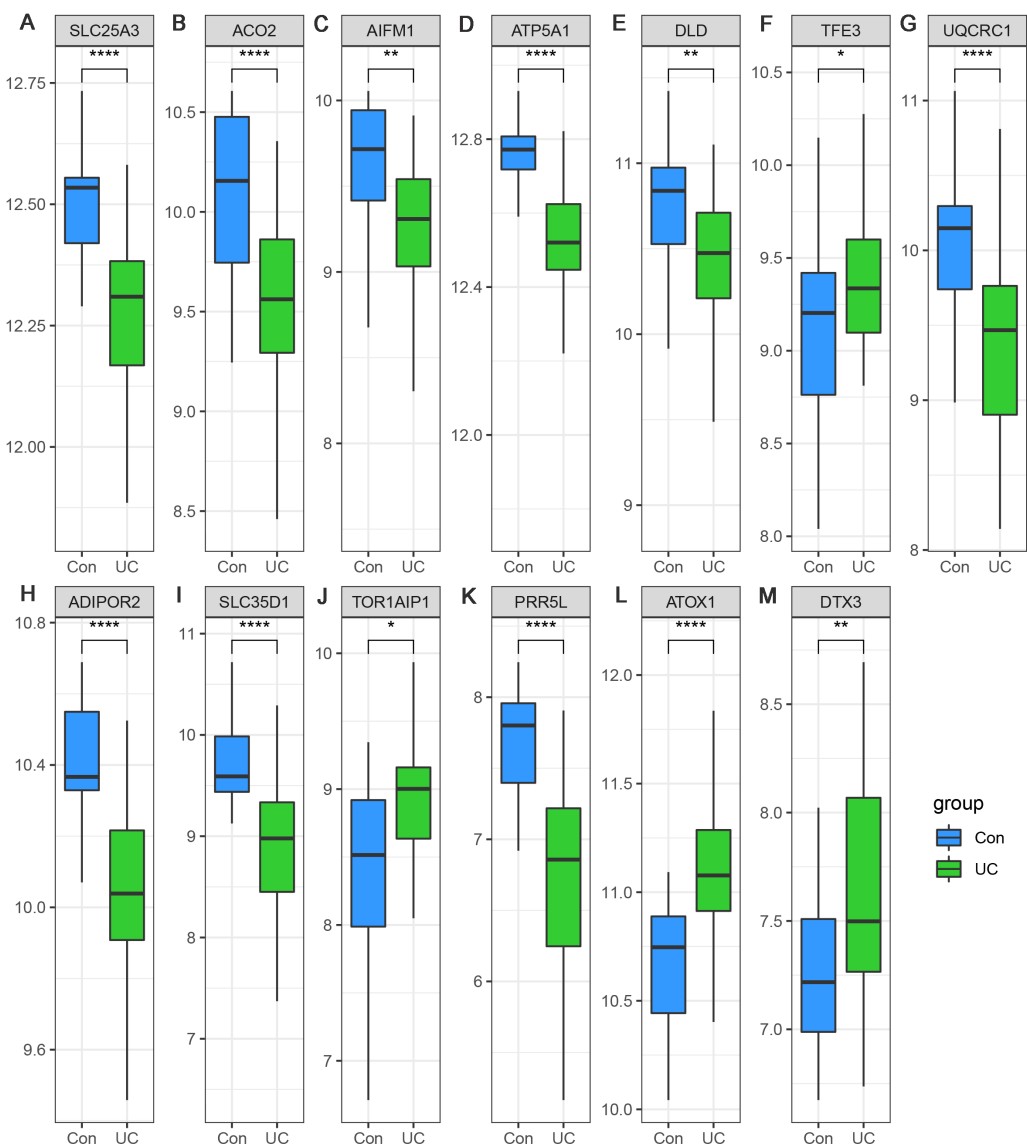

**Figure 6** (A–M) **Validation of the expression change of hub genes in UC from GSE87466.**

*Su & Kadesch, 1990*; *Perera et al., 2015*; *Willett et al., 2017*). In this study, we found that TFE3 was continuously upregulated during UC-associated carcinogenesis. However, the plausible roles of TFE3 in this process remain unclear. Single-gene GSEA revealed that biological processes and KEGG pathways related to angiogenesis and immune response were positively correlated with the upregulation of TFE3 (Figs. 7A, 7C), whereas those related to mitochondrial function and energy metabolism were negatively correlated with the TFE3 upregulation (Figs. 7B, 7D).

**Table 1  Literature review of the proof for the carcinogenic effects of hub genes.**

| Genes | Encoding protein | Known function | Any proof for carcinogenesis | Any proof for CRC |
|---|---|---|---|---|
| SLC25A3 | Phosphate carrier protein, mitochondrial | Transport of phosphate to mitochondrial matrix, involve in ATP synthesis | Yes (*Oehler et al., 2009*) | No |
| ACO2 | Aconitate hydratase, mitochondrial | Catalyze the isomerization of citrate to isocitrate, involve in tricarboxylic acid cycle | Yes (*Ciccarone et al., 2020*; *Wang et al., 2013*) | No (*Laiho et al., 2003*) |
| AIFM1 | Apoptosis-inducing factor 1, mitochondrial | Regulate apoptosis, and involve in mitochondrial respiratory | Yes (*Liu et al., 2018*; *Rao et al., 2019*) | Yes (*Millan & Huerta, 2009*) |
| ATP5A1 | ATP synthase subunit alpha, mitochondrial | A component of the mitochondrial complex V in the respiratory chain, involve in ATP synthesis | Yes (*Bruggemann et al., 2017*) | Yes (*Seth et al., 2009*) |
| DLD | Dihydrolipoyl dehydrogenase, mitochondrial | Function as a dehydrogenase regulating energy metabolism, or as a protease | Yes (*Khalil, 2007*) | No |
| TFE3 | Transcription factor binding to IGHM enhancer 3 | Function as transcription factor, regulate many biological processes, such as energy metabolism, lysosomal biogenesis and immune response | Yes (*Linehan et al., 2019*; *Perera et al., 2015*) | No |
| UQCRC1 | Cytochrome b-c1 complex subunit 1, mitochondrial | A component of the complex III in the respiratory chain, involve in ATP synthesis | Yes (*Wang et al., 2020*) | Yes (*Li et al., 2019*) |
| ADIPOR2 | Adiponectin receptor protein 2 | Regulates glucose and lipid metabolism, mediate increased AMPK and PPAR-alpha ligand activities | Yes (*Chung et al., 2017*) | Yes (*Byeon et al., 2010*) |
| SLC35D1 | UDP-glucuronic acid/UDP-N-acetylgalactosamine transporter | Transport both UDP-glucuronic acid and UDP-N-acetylgalactosamine to ER lumen, participate in glucuronidation and/or chondroitin sulfate biosynthesis | Yes (*Viola et al., 2017*) | No |
| TOR1AIP1 | Torsin-1A-interacting protein 1 | Localize to the inner nuclear membrane, maintain the attachment of the nuclear membrane to nuclear lamina during cell division | No | No |

*(continued on next page)*

**Table 1** (*continued*)

| Genes | Encoding protein | Known function | Any proof for carcinogenesis | Any proof for CRC |
|---|---|---|---|---|
| PRR5L | Proline-rich protein 5-like | Associate with mTORC2 complex that regulate cellular processes including survival and cytoskeleton organization | No (*Thedieck et al., 2007*) | No |
| ATOX1 | Copper transport protein | Bind and deliver cytosolic copper to copper ATPase proteins, and be important in cellular antioxidant defense | Yes (*Karginova et al., 2019*; *Kim et al., 2019*) | Yes (*Jana et al., 2020*) |
| DTX3 | Deltex E3 ubiquitin ligase 3 | Function as an E3 ubiquitin ligase, regulate Notch signaling | Yes (*Ding et al., 2020*) | No |

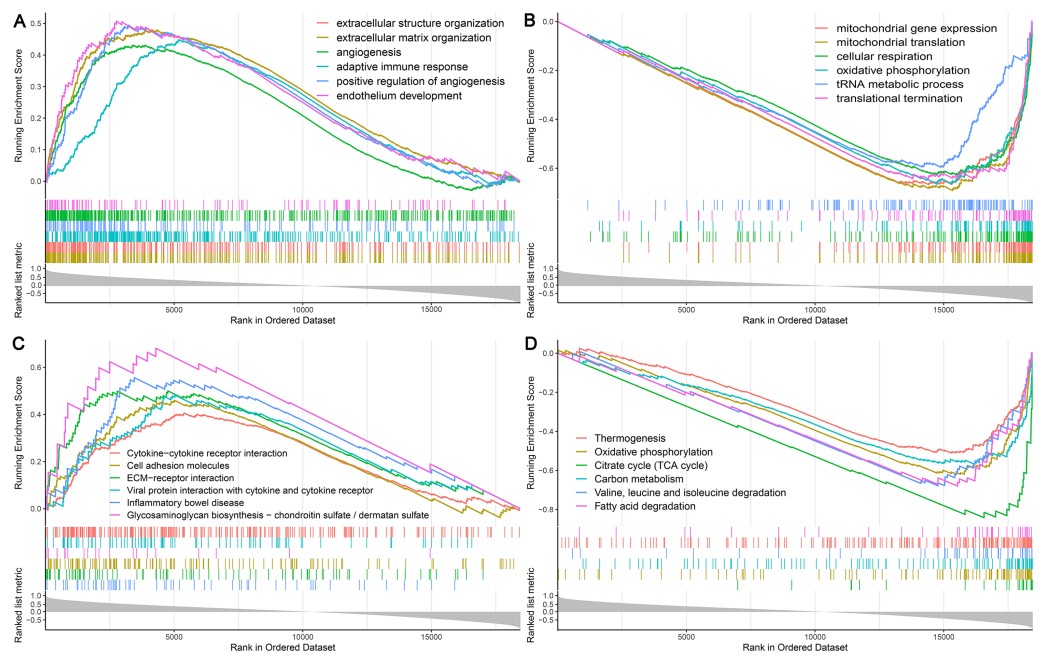

**Figure 7** **Single-gene GSEA analysis of TFE3.** (A) Enriched biological processes of genes positively correlated with TFE3. (B) Enriched biological processes of genes negatively correlated with TFE3. (C) Enriched KEGG pathways positively correlated with TFE3. (D) Enriched KEGG pathways negatively correlated with TFE3.

## DISCUSSION

Most traditional research related to understanding of carcinogenic mechanisms focuses on comparison between tumor and normal tissue; however, this idea seems to be not fully applicable to UC-CRC considering its gradual molecular changes from inflammation to cancer. As UC-CRC arises due to the chronic inflammation of the colonic mucosa, it is logical to find the carcinogenic genes and biological processes associated with inflammation

during the progression of UC (*Itzkowitz & Yio, 2004*). In the present study, we identified two functional modules by WGCNA using the transcriptional profiles of colonic mucosa from healthy individuals and UC and CAD patients, which are correlated with UC and CAD in the same direction, suggesting their potential carcinogenic function when they contribute to inflammation. Thirteen hub genes (*SLC25A3*, *ACO2*, *AIFM1*, *ATP5A1*, *DLD*, *TFE3*, *UQCRC1*, *SLC35D1*, *TOR1AIP1*, *PRR5L*, *ATOX1*, and *DTX3*) were identified, and their continuous up/down-regulation supported their continuous roles during the process of progression from inflammation to carcinogenesis.

Functional enrichment analysis revealed that the green module genes were significantly enriched in biological processes and pathways related to mitochondrial function and energy metabolism. Consistently, almost all the hub genes of the green module (*SLC25A3*, *ACO2*, *ATP5A1*, *DLD*, and *UQCRC1*) were found to have involved in mitochondrial functions, such as the TCA cycle and ATP synthesis. Mitochondria have been discovered to play a multifunctional role in the pathogenesis of cancers (*Sajnani et al., 2017*). The well-known "Warburg effect" describes the shift in energy metabolism of cancer cells from mitochondrial respiration to glycolysis (*Warburg, 1956*). This shift may result from the mutation or dysregulation of genes related to energy metabolism, such as the genes encoding the enzymes involved in the TCA cycle. A decreased copy number of mtDNA has been found in lung cancer, liver cancer, gastric cancer, and CRC (*Lee et al., 2005*). That is to say, the downregulated hub genes related to mitochondrial respiration during UC inflammation may contribute to the transformation from UC to cancer by participating in the shift of energy metabolism. In addition, the alterations in these hub genes may lead to abnormal enzymatic reactions and oncometabolites, which in turn exert carcinogenic effects mainly through epigenetic regulation (*Nowicki & Gottlieb, 2015*). Another hub gene of the green module, *AIFM1*, is a well-known caspase-independent death effector released from mitochondria, and was later discovered to also play a role in oxidative phosphorylation (OXPHOS) by regulating complex I proteins post-transcriptionally. Both cell death and OXPHOS processes have been involved in cancer pathogenesis (*Liu et al., 2018*; *Rao et al., 2019*), but the role of *AIFM1* during UC-associated carcinogenesis needs further investigation.

TFE3 is a member of the MiT family of helix-loop-helix leucine zipper transcription factors that regulate their target genes by binding to E-box sequences in promoters. E-box sequences are mainly found in genes related to energy metabolism, including those involved in glycolysis and lipid metabolism. TFE3 has been reported to regulate mitochondrial dynamics and function in the liver (*Pastore et al., 2017*). Moreover, TFE3 has been validated as an oncogene in kidney and pancreatic cancers by virtue of its regulation of metabolic and non-metabolic pathways (*Linehan et al., 2019*; *Perera et al., 2015*). In the present study, single-gene GSEA revealed a link between TFE3 and mitochondrial function, while the latter was considered to be highly involved in UC-associated carcinogenesis. Therefore, it is logical to speculate that TFE3 may play a role in UC-associated carcinogenesis by regulating mitochondrial function. Further experimentation is required to confirm this hypothesis.

The genes encoding for proteins involved in cell–cell junctions and immune responses along with those involved in pathways, including the Hippo signaling pathway, Notch

signaling pathway, and ErbB signaling pathway, were significantly enriched in the blue module. Cell–cell junctions are vital for tissue homeostasis as they not only maintain the barrier function, but regulate complex cellular signaling networks related to cell proliferation and migration (*Garcia, Nelson & Chavez, 2018*). Disruption of tight junctions and subsequent immune dysfunction due to exposure to antigens play critical roles in UC pathogenesis. The Hippo pathway is an important signaling pathway regulating cell proliferation and apoptosis, and its dysregulation contributes to cancers. Moreover, the activity of the Hippo pathway is highly dependent on cell junctions (*Karaman & Halder, 2018*). Therefore, it is logical to think that the altered cell junctions and relevant signaling network during the inflammatory phase of UC may contribute to carcinogenesis (*Feigin & Muthuswamy, 2009*).

The obvious limitation of the present study is the small sample size; however, additional CAD or UC-CRC clinical specimens were not available for validation. The same expression trend of hub genes in another UC dataset and the reported relationships between these genes and cancers may help in demonstrating the reliability of this study. Further efforts are being made in this direction. This work is expected to provide new insights into the process of UC-associated carcinogenesis.

## CONCLUSIONS

In conclusion, this study, using WGCNA, found two gene modules that were significantly correlated with the process of UC-associated carcinogenesis from inflammation to dysplasia. From these two modules, 13 hub genes (*SLC25A3*, *ACO2*, *AIFM1*, *ATP5A1*, *DLD*, *TFE3*, *UQCRC1*, *ADIPOR2*, *SLC35D1*, *TOR1AIP1*, *PRR5L*, *ATOX1*, and *DTX3*) were identified as key genes involved in UC-associated carcinogenesis. Functional enrichment analysis revealed the critical biological processes contributing to UC-associated carcinogenesis, mainly include mitochondrial dysfunction, cell–cell junction, and immune responses. TFE3, a transcription factor related to mitochondrial function and cancer, seem to play a central role in this process.

## ACKNOWLEDGEMENTS

We would like to thank Editage for English language editing.

### Funding

This work was supported by the Chinese Academy of Medical Sciences (CAMS) Initiative for Innovative Medicine (CAMS-2016-I2M-1-007). The funders had no role in study design, data collection and analysis, decision to publish, or preparation of the manuscript.

### Grant Disclosures

The following grant information was disclosed by the authors:
Chinese Academy of Medical Sciences (CAMS) Initiative for Innovative Medicine: CAMS-2016-I2M-1-007.

## Competing Interests

The authors declare there are no competing interests.

## Author Contributions

- Di Zhang conceived and designed the experiments, performed the experiments, analyzed the data, prepared figures and/or tables, authored or reviewed drafts of the paper, and approved the final draft.
- Pengguang Yan performed the experiments, analyzed the data, prepared figures and/or tables, and approved the final draft.
- Taotao Han analyzed the data, prepared figures and/or tables, and approved the final draft.
- Xiaoyun Cheng analyzed the data, authored or reviewed drafts of the paper, and approved the final draft.
- Jingnan Li conceived and designed the experiments, authored or reviewed drafts of the paper, and approved the final draft.

## Data Availability

The code is available in the Supplementary File. The data for analysis and validation are available at NCBI GEO: GSE47908 and GSE87466.

## Supplemental Information

Supplemental information for this article can be found online at http://dx.doi.org/10.7717/peerj.11321#supplemental-information.

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
