# Peer review of "Identification of key genes and biological processes contributing to colitis associated dysplasia in ulcerative colitis"

_PeerJ, doi:10.7717/peerj.11321_

## Round 0.1 · original submission · Major Revisions

Your manuscript was considered interesting and valuable by the reviewers, but one of the reviewers raised some important comments that need to be addressed. Specifically, the reviewer suggested that you perform additional comparisons, that will help better analyze the different steps of cancer progression and quantify the differences between UC and CAD. They also suggested that you revise the introduction to include transcriptomic, proteomic and genetic studies performed on other cancers, such as breast.

Please, submit a detailed rebuttal which shows where and how you have taken all comments and suggestions into consideration. If you do not agree with some of the reviewers’ comments or suggestions, please explain why. Your rebuttal will be critical in making a final decision on your manuscript. Please, note also that your revised version may enter a new round of review by the same or by different reviewers. Therefore, I cannot guarantee that your manuscript will eventually be accepted.

Reviewer 1 ·

Basic reporting

Major comments:
1. In the introduction, the authors should replace their study in the context of all the studies looking for early genetic changes contributing to cancer progression. For example, the characterization of premalignant lesions has been conducted in the context of breast cancer (See Ju et al in Cancers 2020) and it was shown that 70% of miRNA alterations occur during the initial progression from normal to a preneoplastic stage. This result is even more worth to mention as it support the results shown later by the authors.

2. In the differential expression analysis, the authors wrote that they only compared CAD to control but in figure 2 show UC compared to control. That brings me to a comment that apply to the manuscript overall, the authors should make 3 clear comparisons that reflect the steps of cancer progression:
- control vs UC,
- control vs CAD,
- UC vs CAD.
I think they have the dataset and the opportunity to detail the genetic changes happening early in the carcinogenesis if they show these comparisons. That will create also more continuity in their manuscript as they used the 3 groups again in their hub genes analyses. Furthermore, it will be a great addition to quantify how similar are UC and CAD samples overall. How many genes show an increase in expression only in UC or CAD? Venn diagram are a useful tool to present that type of data.

3. Even if the dataset used does not contain a CRC group. It will be easy to look for the genes they identified in another CRC dataset. Rather than a review of the literature, I think that will serve better their purpose of demonstrating that the identified genes play a role in carcinogenesis.

4. Their explanation of the correlation between the modules and the progression of UC to dysplasia is a little confusing, they should rephrase and detailed more. What do they define as “progress” in Figure 3C for example?


Minor comments

1. Could the authors pool the two datasets to increase statistical significance as they showed that they give similar results? Notably, that will increase their chances to identify differences between UC, CAD and controls.

2. It is not clear how the gene significance threshold and the module membership cut-off were determined. If there is a reference, please cite it.

3. The English language should be improved to ensure that an international audience can clearly understand the manuscript. Some examples where the language could be improved are the abstract and lines 54, 58, 67, 194, 201 – the current phrasing makes comprehension difficult.

4. Figures 1B and C and figures 5B, D: Authors should keep the same color for each group across the figures.

5. Figure 2B: It is impossible to read the Genes names, please provide a figure with a better definition.

Experimental design

No comment

Validity of the findings

No comment

Additional comments

In this manuscript, the authors proposed an interesting study based on data mining to look for the genes and biological processes involved in the transition from inflamed to premalignant lesions and ultimately to colorectal cancer. They have realized a substantial work of biostatistical analyses and the genes they identified could stimulate further experimental studies. However, I believe that there is a lot of room from improvement of the current manuscript. I detail the major issues I found below, as well as some other minor concerns, and suggest some ways to improve the manuscript.

·

Basic reporting

No comment

Experimental design

The sample size is small to jump into a conclusion and warrant further explanation and to also recommend for further study with larger sample size . Details of UC patients such as age, smoking status, intensity of colonic inflammation (based on endoscopic score), disease duration, concomitant medications taken (any potential chemoprophylaxis such as mesalamine), co-existence of primary sclerosing cholangitis,family history of CRC or UC are important factors towards UC-CAD and neoplasia. These parameters need to be mentioned in the manuscript.

Validity of the findings

Mitochondrial dysfunction that potentially lead to UC-CRC warrant further explanation in this study in the context of different disease activity and disease duration. I recommend also to compare with CRC arises from the polyp (sporadic CRC) in terms of mitochondrial dysfunction issue.

Additional comments

Overall is good but the sample size is very small and the validation is not determined. Need to take into account that sporadic CRC occurs slightly younger age group in Asian population and UC is low prevalent too in Asian countries. Hence, the possible pick rate of UC-CRC in the low prevalent area is very low and it might not be cost-effective and useful in daily clinical practice.

---

## Round 0.2 · Minor Revisions

You addressed all the reviewers' comments satisfactorily, however I have a couple of very minor comments that need to be addressed prior to your manuscript being accepted.

First, there are still some minor issues with the English language in the manuscript. For example see "founded" (page 4, line 101), "offers possibility" (page 4, line 106), "Sing-gene GSEA" (page 7, line 365), "from health UC" (page 8, line 404).

Secondly, one of the reviewers asked you to clarify the phenotype "progress", which you did in the materials and methods section. You use the term "progress" in the results subsection of the abstract (page 2, line 33). This is confusing so please clarify.

Reviewer 1 ·

Basic reporting

No comment

Experimental design

No comment

Validity of the findings

No comment

Additional comments

The authors made major revisions to their article and I am satisfied with their answers. I do not have more comments.

---

## Round 0.3 · accepted · Accept

Thank you for implementing all the suggestions. The manuscript is greatly improved.